# *Streptococcus equi* Subspecies *zooepidemicus* Endocarditis and Meningitis in a 62-Year-Old Horse Rider Patient: A Case Report and Literature Review

**DOI:** 10.3390/microorganisms12112201

**Published:** 2024-10-31

**Authors:** Giacomo Franceschi, Alessandra Soffritti, Matteo Mantovani, Margherita Digaetano, Federica Prandini, Mario Sarti, Andrea Bedini, Marianna Meschiari, Cristina Mussini

**Affiliations:** 1Department of Infectious Diseases, Azienda Ospedaliera-Universitaria of Modena, University of Modena and Reggio Emilia, 41125 Modena, Italy; ale.soffritti@gmail.com (A.S.); andreabedini@yahoo.com (A.B.); mariannameschiari1209@gmail.com (M.M.); 2Clinical Microbiology Laboratory, Azienda Ospedaliera-Universitaria of Modena, University of Modena and Reggio Emilia, 41125 Modena, Italy

**Keywords:** *Streptococcus equi*, zoonotic infection, infective endocarditis, pneumonia, meningitis, antibiotic therapy, zoonosis, aortic valve replacement

## Abstract

The present article presents a case report and literature review concerning the *Streptococcus equi* subspecies *zooepidemicus* (SEZ), a rare zoonotic pathogen in humans. The case involves a 62-year-old man with no prior heart disease, presenting with endocarditis, pneumonia, and meningitis following close contact with a horse. The patient underwent urgent aortic valve replacement due to severe valvular damage caused by the infection. Blood and cerebrospinal fluid cultures confirmed the presence of SEZ, and the patient was treated with a combination of antibiotics, followed by a successful step-down to oral therapy using linezolid. A review of 25 additional *Streptococcus equi* endocarditis cases highlights the rarity of the condition, its association with animal contact, and its tendency to cause multi-site infections, such as pneumonia and meningitis. Early diagnosis, appropriate antibiotic therapy, and, in severe cases, surgical intervention are critical for a favorable outcome. This report emphasizes the importance of recognizing zoonotic infections in at-risk populations and the potential need for public health surveillance in these scenarios.

## 1. Introduction

*Streptococcus equi* (*S. equi*) is a Gram-positive non-spore-forming coccus, which is part of the pyogenic streptococci group. Based on the hemolysis patterns on blood agar plates, it is classified as a β-hemolytic (complete lysis) bacterium and can be further differentiated by carbohydrate group antigens of Lancefield group C. 

Three different subspecies have been recognized: *S. equi* subsp. *equi* (*S. epidemicus* (SEE)), S. *equi* subsp. *zooepidemicus* (*S. zooepidemicus* (SEZ)), and *S. equi* subsp. *ruminatorum* (S. *ruminatorum* (SER)).

Among the subspecies of *S. equi*, SEE is primarily a pathogen of horses and is responsible for strangles, a highly contagious infection of the upper respiratory tract associated with the lymph nodes of solipeds [1].

SER was identified for the first time in 2004, isolated from mastitis in small ruminants [2]. Although only a few cases have been reported in the literature, SER has the potential to cause zoonotic infections in humans [3].

SEZ was first isolated in 1934 by P. R. Edwards and was initially named animal pyogens A [4]. It is a part of the mucosal flora in horses and is considered an opportunistic pathogen. Although usually a harmless commensal, *S. zooepidemicus* can occasionally become an important cause of respiratory disease and metritis in horses and cause a wide variety of infections in pigs, sheep, cows, goats, and several other mammalian species [5,6,7].

Transmission to humans is rare and is related to close contacts with horses [8] and the consumption of unpasteurized dairy products [9,10], goat cheese [11], or pork [12].

Between November 2021 and May 2022, Italy experienced an outbreak of SEZ infection, involving a total of thirty-seven clinical cases in the central region of the country, resulting in the death of five patients due to meningitis [13].

Additionally, other severe infections, such as infective endocarditis (IE) [10], septic arthrtis [14] and pneumonia [15], and post-streptococcal complications, such as glomerulonephritis [16] and rheumatic fever [8], have also been identified.

IE is a severe disease that is associated with high morbidity and mortality rates, and according to the latest clinical guidelines [17,18], the diagnosis of IE relies on a combination of clinical, microbiological, and imaging criteria, with the modified Duke criteria being the standard diagnostic framework.

In 2009, the International Collaboration on Endocarditis Prospective Cohort Study (ICE-PCS) showed that the most frequent microorganisms causing IE were *S. aureus* (31%), followed by the viridans group streptococci (17%) and Coagulase-Negative Staphylococci (11%) [19]. Similar results were reported in the EURO-ENDO registry [20].

IE, due to group A, B, C, or G streptococci, including the *Streptococcus anginosus* group (*S. constellatus*, *S. anginosus*, and *S. intermedius*) is relatively rare [21,22]. The previous analysis of 88 patients with group C Streptococcal bloodstream infection (BSI) revealed that IE was the most frequently reported clinical manifestation, occurring in 27.3% of the cases, and SER was isolated in 17.1% of whole cohort with BSI [23]. Mortality was high (25.0%), especially among the older patients and the patients with endocarditis, meningitis, and disseminated infection. Indeed, group C streptococcal bacteremia does not differ from BSI caused by other β-hemolytic streptococci in terms of clinical presentation, treatment, or outcome. 

Among streptococci IE infections, the most common organisms belong to the viridans group (VGS) [19]. β-hemolytic streptococci (BHS) are recognized for their pathogenic role mainly in skin and soft tissue infections and are an uncommon cause of IE. Fernández Hidalgo [21] studied the data from the ICE-PCS cohort; among 4794 cases of definite IE, 1336 (27,9%) cases were caused by streptococci, of which 823 (61,6%) were caused by VGS, and 147 (11%) by BHS. S. agalactiae (group B) was the most common pathogen among the BHS cases. Overall, IE caused by BHS represented 3.1% of all the IE episodes. These findings show that BHS IE is an aggressive disease characterized by an acute presentation and higher rates of stroke, systemic emboli, and in-hospital mortality than those of the viridans group streptococci IE. 

One of the primary challenges is the lack of comprehensive data on the clinical presentation, therapy, and outcomes of *S. equi*-related IE. 

Our goal is to address these gaps by presenting a detailed case report of an immunocompetent patient and conducting a literature review to summarize the clinical characteristics, treatment approaches, and outcomes of this specific zoonotic infection.

## 2. Materials and Methods

We conducted a literature review of the cases of infective endocarditis caused by *S. equi* spp. 

Bibliographic research was carried out on the PubMed, ScienceDirect, Google Scholar, ResearchGate, and Jstor Databases on 1 September 2024 by seeking combinations of the following keywords: “Streptococcus equi” and “Endocarditis”. We included articles written in all languages and excluded case reports with insufficient key elements, such as detailed clinical characteristics, demographic data, therapeutic approaches, or patient outcomes. Table 1 reports the clinical cases.

Based on the above-mentioned criteria, we selected 21 articles describing 24 patients with endocarditis caused by *S. equi*. Including our patient, a total of 25 patients were counted in analyses. The first study was published in 1982, and the last one in September 2023. *S. equi* endocarditis was reported in 12 countries: 11 from North and South America, 12 from Europe, 1 from Australia, and 1 from Turkey. Twenty studies described only 1 case, while one study described 4 cases. Seventeen articles were written in English, three in Spanish, and one in Turkish. Three articles were posters, while eighteen were published in scientific journals.

## 3. Case Report Description

On 12 May 2024, a 62-year-old male presented to the emergency department in a tertiary care hospital in Central–Northern Italy, with intermittent high fever, a cough, and reporting memory deficits and confusion. 

Past medical history included hypertension and hypercholesterolemia. His symptoms started two weeks before and had worsened despite taking self-administered medications. 

Initial laboratory investigation showed a white blood cell (WBC) count of 16,730/mm^3^ (92% neutrophils); the C-reactive protein concentration (CRP) was 23.3 mg/dL (normal < 0.7 mg/dL). 

After brain computed tomography (CT), which showed no parenchymal lesions of hemorrhagic or ischemic origin, lumbar puncture was performed. The cerebrospinal fluid (CSF) obtained was slightly turbid, with a WBC count of 351/mm^3^ and with clear neutrophil prevalence, a glucose level of 31 mg/dL, and a protein level of 145 mg/dL (normal 20 mg/dL to 50 mg/dL). Then, empiric therapy for bacterial meningitis was started, including 2 g IV ceftriaxone every 12 h, 3 g IV Ampicillin every 8 h, and 10 g IV dexamethasone every 6 h, pending the results of Gram stain and culture. PCR FilmArray performed on CSF was negative. 

Two days after admission, the patient was transferred to the intensive care unit due to acute pulmonary edema and consequent worsening respiratory dynamics and underwent intubation and mechanical ventilation. Levofloxacin 750 mg was empirically administered for pneumonia (Figure 1).

The same day, transthoracic and transesophageal echocardiograms (TTEs and TOEs) showed large-scale vegetation on the aortic valve (8.63 mm × 18.48 mm), with concomitant severe aortic insufficiency and cusp perforation consistent with the diagnosis of IE (Figure 2). 

Blood cultures drawn on admission reported Gram-positive coccus chain growth in 9 h. Matrix-assisted laser desorption ionization–time-of-flight mass spectrometry (MALDI-TOF MS) was performed, and SEZ was identified in both the CSF and aerobic and anaerobic blood cultures bottles.

The pathogen was susceptible to β-lactams, fluoroquinolones, glycopeptides, linezolid, and rifampin, but was resistant to macrolides, tetracyclin, co-trimoxazole, and clindamycin. The follow-up blood cultures obtained after 48 h of treatment were negative.

Antibiotic therapy was tailored according to susceptibility testing to 3 g cefotaxime every 6 h and 5 mg/kg/day gentamicin. 

On day 2 of hospitalization, the patient underwent urgent aortic valve replacement with a 23 mm Edwards Resilia biological prosthesis. Recovery after cardiac surgery was uneventful, and valve culture remained negative.

Cefotaxime was continued at 3 g four times a day, and gentamicin was discontinued after 2 weeks. 

The thoraco-abdominal CT scan with contrast performed on day 17 post admission showed normal findings post-aortic valve replacement, no further valve vegetation, and no focal lesions in the liver, spleen, pancreas, kidneys, or adrenals.

Follow-up TOE showed a well-functioning prosthetic valve and preserved systolic function, with an ejection fraction of 55%.

Thus, the patient was discharged 19 days after admission; antibiotic therapy was administered orally with 600 mg linezolid every 12 h for an additional two weeks.

The patient reported owning a horse, and upon further questioning, revealed that he had administered aerosol therapy on his horse for upper respiratory symptoms some weeks before admission.

Therefore, the case was reported to the veterinary public health surveillance service for further investigation. No additional regulatory actions were required, as the infection is not classified as a transmissible animal disease according to the “Animal Health Law” (Regulation (EU) 2016/429) [41].

## 4. Literature Review

### 4.1. Demographic and Clinical Characteristics of the Cases Reviewed

Table 2 summarizes the demographic characteristics of the 25 patients with IE, the amount of multifocal infections, the valves affected, the diagnostic approaches, the antibiotic therapy (ABT) administrated, the frequency of heart valve replacement, their outcome of acute infection, and the presence of long-term injury. 

Nineteen (76.0%) patients were male, with a mean age of 65 years (IQR 57–73). IE was caused by SEZ in twenty (88.0%) cases and by SER in two (8.0%) cases, while in one case (4.0%), the subspecies could not be established. We analyzed the source of infection among the case reports. 

On admission, the patients had an average C-reactive protein (CRP) level of 279.5 mg/L and an average white blood cell (WBC) count of 13.5 × 10^3^ cells/mm^3.^

All the patients (*n* = 25, 100%) had positive blood cultures for *S. equi*. However, only one patient had a positive valve culture, while another patient had a negative valve culture, but was positive in molecular testing.

### 4.2. Risk Factors and Comorbidities

The analysis of potential risk factors revealed that fourteen patients (60.0%) had contact (direct or undirect) with horses, while in seven cases (28.0%), the consumption of unpasteurized milk or dairy products was not reported as the main risk factor for transmission. In three cases, it was not possible to identify any potential infectious source.

The cohort exhibited a moderate level of comorbidity, with a mean Charlson Comorbidity Index (CCI) of 2.72 [42].

Nine patients (36.0%) suffered from hypertension, but otherwise showed a low prevalence for other pre-existing diseases, such as diabetes mellitus (n = 2, 8.0%) or chronic kidney disease (n = 2, 8.0). 

On the other hand, in almost one third of the patients (n = 8, 32.0%), a high-risk condition was observed, such as prosthetic valve replacement [43,44] (n = 6, 24.0%), and other predisposing factors, like congenital valve anomalies (n = 1, 4.0%) and rheumatic heart disease (n = 1, 4%) [45,46].

### 4.3. Multifocal Infections and Valve Involvement

Sixteen (64.0%) patients had multi-site infections, with at least one organ affected. Pneumonia (n = 9, 36.0%), meningitis (n = 7, 28.0%), arthritis (n = 3, 12.0%) and spondylodiscitis (n = 3, 12.0%) were the most common multifocal infections.

Regarding the clinical features of *S. equi* spp. IE, the aortic valve was affected in fifteen (60.0%) cases, and the mitral valve in six (24.0%). Septic emboli were present in 14 (56.0%) patients.

### 4.4. Antibiotic Therapy and Outcomes

Combination ABT was the most common treatment approach (n = 18, 72%), and Benzylpenicillin (PenG) was the most commonly prescribed ABT in 26.8% of the total prescriptions, followed by a third-generation cephalosporin (CEP) in 24.3%.

Concerning simultaneous antibiotic administration, gentamicin was prescribed a total of ten times (24.3%), and half of the prescriptions (n = 5, 50.0%) were given in combination with PenG.

The median duration of antibiotic treatment was 6 weeks, with a range of 4–16 weeks. Valve replacement surgery was performed in eight (32.0%) cases.

The mortality rate was 20%, while six (24.0%) survivors had residual injuries, and nine (36.0%) survived without any sequelae.

## 5. Discussion

To the best of our knowledge, this is the first *S. equi* spp. IE literature review.

When comparing the characteristics of our study population with the Danish cohort studied by Chamat-Hedemand [47], which investigated the prevalence of infective endocarditis in 6506 cases of streptococcal bloodstream infections (BSIs), our sample, although significantly smaller, shows a slightly lower mean age (65.5 vs. 68.1 years).

Furthermore, *S. equi* spp. IE compared to that in the Danish study exhibits a marked male predominance (76.0% vs. 52.8%). This pronounced difference is likely due to occupational exposure, as men are more frequently involved in activities that involve contact with horses, such as those occurring in stables or rural settings.

Indeed, our literature review, as well as our case report, clearly indicates that this zoonosis is closely associated with contact with animals (predominantly livestock) or the consumption of unpasteurized dairy products.

Nearly all IE cases caused by *S. equi* are attributable to SEZ, consistent with the literature reports highlighting the greater pathogenicity of this subspecies compared to that of others due to several virulence factors that contribute to invasive diseases [10].

Another aspect to highlight in our literature review is that the patients had a moderately average Charlson Comorbidity Index and few other pre-existing conditions, indicating that they were mostly healthy individuals. 

However, it is clear that *S. equi* IE follows the typical pathogenesis of endocarditis; a significant portion of the subjects had pre-existing valvular abnormalities, which represent locus minoris resistentiae and facilitate the adherence of *S. equi* bacteremia to the endothelial surface [20].

Furthermore, our literature review demonstrates that S. equi, similar to other Gram-positive cocci, such as *S. aureus* [48] and *S. gallolyticus* [49], possesses a wide array of virulence and adhesiveness mechanisms. In fact, approximately two out of three patients in our review presented with concomitant *S. equi* infection in another organ in addition to endocarditis. 

Regardless of the portal of entry, pneumonia is the most frequently observed multifocal infection in cases of *S. equi* IE. Moreover, consistent with the other literature data regard BHS IE [21], *S. equi* shares with them greatly pathogenic characteristics, and septic emboli were observed in more than half of our study populations. The excessive rates of systemic embolization and congestive heart failure in BHS IE suggests that early surgery may be important to prevent the progression of disease [21].

In 12 out of 25 cases, diagnosis was made using TTE. However, it is important to note that in three cases, which represents 20% of the total TTE incidences in our literature review, were negative. In these instances of the strong suspicion of IE, diagnosis was ultimately confirmed through TOE. 

According to the literature, TTE demonstrates low sensitivity, but good specificity as compared with those of TOE when evaluating IE, particularly in cases involving prosthetic valves [47,50,51]. As recommended in the 2023 ESC Guidelines [51], our findings confirm the role of TOE when the TTE results are inconclusive and in patients with a negative TTE and a strong suspicion of IE. 

The antibiotic treatment of BHS IE is similar to that of oral streptococci, except that short-term 2-week therapy is not recommended, and gentamicin should be given for 2 weeks. The treatment in our literature review aligned with the 2023 ESC Guidelines [18] as PenG and third-generation CEP were the primary antibiotics prescribed in the case reports.

In our case, the strain was fully susceptible to penicillin, but given concomitant meningitis, PenG was avoided because of its poor penetration into the CSF, and we replaced it with third-generation CEP [52]. Gentamicin was administered for two weeks to synergize with the cell wall inhibitors for bactericidal activity. Surgery contributed by removing the infected material.

In adherence with the POET trial [53], our strategy involved step-down outpatient oral antibiotic treatment once the patient was clinically stable and the follow-up TOE was negative.

This case report represents the first documented instance of *S. equi* IE successfully treated with step-down oral therapy using linezolid tablets. 

This choice was determined due to its status as one of the few oral agents fully effective against both the strains isolated from the CSF and blood.

Furthermore, our choice of linezolid was based on its pharmacokinetic/pharmacodynamic properties. Although it is a bacteriostatic antibiotic, its small molecular size and moderate lipophilicity allow for good tissue distribution [54]. Considering the multifocal dissemination described in our case report, our goal was to maintain adequate drug concentrations in all the tissues throughout the entire duration of antibiotic therapy.

Our strategy was effective; the infection was effectively treated, and the follow-up visits revealed the complete eradication of infection. The treatment successfully cured IE, pneumonia, and meningitis.

## 6. Conclusions

Of all the group C streptococcal subspecies, SEZ is probably the most aggressive human pathogen, and several outbreaks and sporadic cases of severe infection due to this microorganism have been reported. In most cases, SEZ leads to fulminant infection. 

SEZ represents a significant zoonotic threat, particularly for individuals who undergo occupational or environmental exposure to horses or unpasteurized dairy products. This case report highlights the importance of timely diagnosis and treatment to prevent severe outcomes, such as multi-site infections and valvular damage. Early recognition and the appropriate use of antibiotics, along with surgical intervention when necessary, are crucial for patients’ recovery. 

Finally, our case report suggests that linezolid is an effective option for oral switch therapy after initial intravenous treatment in streptococcal IE, providing adequate coverage and tissue penetration in the case of multifocal infections.

Further studies are needed to improve our understanding of the clinical management of such rare, but serious infections.

## Figures and Tables

**Figure 1 microorganisms-12-02201-f001:**
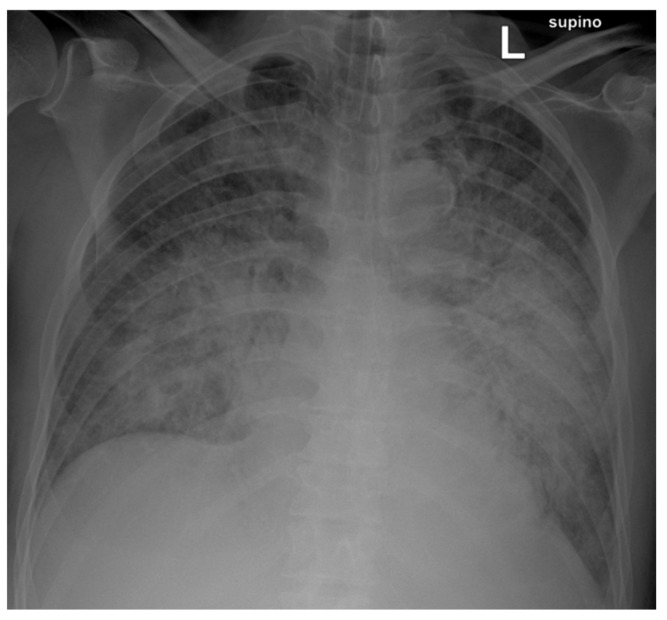
Chest X-ray shows multiple bilaterally diffused areas of predominantly perihilar interstitial–alveolar involvement, with near-complete opacification of both lungs.

**Figure 2 microorganisms-12-02201-f002:**
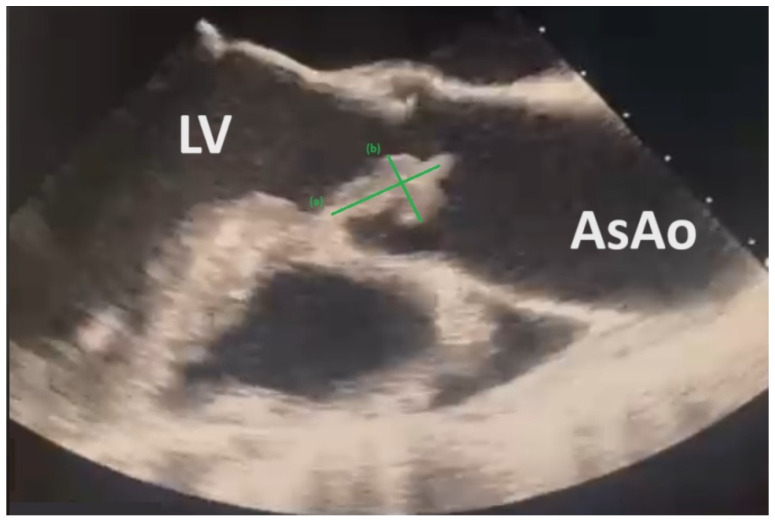
TTE reveals mobile vegetation on aortic valve measuring (**a**) 8.63 mm and (**b**) 18.48 mm. Left Ventricle (LV), Ascending Aorta (AsAo).

**Table 1 microorganisms-12-02201-t001:** Published cases of infective endocarditis due to *Streptococcus equi* spp.

Author (Year)	Age (y.)	Sex	Subspecies	Source of Infection	Previous Heart Diseases	CCI	Multi Focal Infection	Heart Valve	Septic Emboli	TTE	TOE	Antibiotic Therapy	Duration Therapy (wks)	Valve Replacement	Outcome
Franceschi et al. (2024) [this case]	58	♂	*Zooepidemicus*	Contact with horses	No	1	Pneumonia, Meningitis	Aortic	No	Positive	Positive	AMP + CRO ⇒ CTX + GNT ⇒ oral LNZ	5	Yes	Survive, episode of atrial fibrillation
Ali Akram et al. [24] (2023)	85	♀	*Zooepidemicus*	Contact with horses	No	5	No	Aortic	No	Negative	Positive	CRO	4	No	Survive
Saray Ricardo et al. [25] (2022)	69	♂	not determinable	Consumption of unpasteurized milk	No	2	No	Mitral	cerebral ischemic lesion	Positive	np	PIP/TZB ⇒ PEN G	nr	No	Death
Elde et al. [26] (2021)	57	♂	*Zooepidemicus*	walking his dog on trails also used for equestrian activities	No	1	Spondylodiscitis	Aortic	subarachnoid hemorrhage, mycotic aortic aneurysm	Positive	np	VAN + PIP/TZB ⇒ PEN G	6	Yes	Survive, replacement aortic and mitral valve for recurrent endocarditis.
Gilbert et al. [27] (2021)	28	♂	*Zooepidemicus*	Contact with horses	prosthetic valve in CHD	0	No	Aortic	No	nr	nr	VAN + CRO ⇒ PEN G + GNT	nr	Yes	Survive
Garcia et al. [28] (2020)	54	♂	*Zooepidemicus*	Contact with goats and horses	No	1	Pneumonia, Meningitis, Spondylodiscitis	Aortic	spondylodiscitis L2-L3	nr	Positive	CRO	6	Yes	Survive
Sleutjens et al. [29] (2019)	62	♂	*Zooepidemicus*	Contact with horses	mechanical aortic valve	2	No	Aortic	No	Positive	np	PEN G + GNT	6	No	Survive
Høyer-Nielsen et al. [15] (2018)	82	♂	*Zooepidemicus*	Contact with horses	ischemic heart disease, AF	8	Septic arthritis of the shoulder joint	Aortic	No	Positive	Positive	CFX ⇒ PEN G + GNT ⇒ oral AMX +RIF	12	No	Survive
Chang et al. [30] (2018)	75	♂	*Zooepidemicus*	Not found	No	5	Pneumonia, Meningitis, Endophthalmitis	Aortic	hemorrhagic retinitis	Positive	np	CRO + RIF + Intravitreal VAN + CFZ	nr	No	Death
Kutlu et al. [31] (2018)	65	♂	*Zooepidemicus*	Consumption of unpasteurized products	prosthetic aortic valve and coronary bypass	2	Pneumonia, septic arthritis of knee	Aortic	cerebral and paravertebral abscess	Positive	np	AMP/SLB ⇒ AMP + RIF	nr	Yes	Survive
Sargsyan et al. [32] (2017)	61	♀	*Zooepidemicus*	Contact with horses	bioprosthetic mitral valve	2	Pneumonia	Mitral	Janeway lesions	Positive	np	CRO	6	No	Survive
Redondo Calvoa et al. [33] (2016)	77	♂	*Zooepidemicus*	Contact with horses	ischemic heart disease	4	Pneumonia	Aortic	mycotic aneurysm infrarenal	Positive	np	AMX/CLV ⇒ AMP + GNT	nr	No	Survive
Daubié et al. [34] (2014)	63	♂	*Ruminatorum*	Contact with horses	mechanical aortic valve	2	No	Aortic	No	Negative	Positive	AMX iv + GNT	5	Yes	Survive, persistent Atrio ventricular block (PM)
Villamil et al. [35] (2015)	73	♂	*Zooepidemicus*	Contact with horses	mechanical aortic valve	3	No	Aortic	No	Negative	Positive	PEN G + GNT	6	No	Survive
Bîrluţiu et al. [36] (2013)	55	♀	*Zooepidemicus*	Not found	No	1	Meningitis	Aortic	Ischemic stroke	nr	Positive	AMP + CRO ⇒ PEN G + GNT	5	No	Survive
Pelkonen et al. (patient1) [8] (2013)	57	♂	*Zooepidemicus*	Contact with horses	bicuspid aortic valve	1	Meningitis	Aortic	No	nr	nr	PEN G + GNT	5	Yes	Survive
Meyer et al. [37] (2011)	70	♂	*Ruminatorum*	Contact with horses	No	3	Spondylodiscitis	Mitral	Spondylodiscitis	Positive	np	AMX iv + RIF + GNT	16	No	Survive
Poulin et al. [38] (2009)	59	♀	*Zooepidemicus*	Not found	ostium primum atrial septal defect	5	Meningitis, Endophthalmitis	Mitral	Cerebral, splenic and renal emboli	nr	Positive	CRO + RIF+ Intravitreal VAN + CRO	6	Yes	Survive, permanent total blindness due to retinal detachments
Bordes-Benítez et al. (case 8) [10] (2006)	70	♀	*Zooepidemicus*	consumption of fresh cheese (“queso fresco”)	No	3	Pneumonia	nr	Not reported	nr	nr	Β-lactam agent	nr	nr	Survive, disorientation to time and place
Lee et al. [39] (2004)	79	♂	*Zooepidemicus*	Contact with fresh horse manure for his garden	No	3	Meningitis, bilateral septic arthritis of knees	Aortic	No	nr	Positive	PEN G + VAN	6	No	Survive
Edwards et al. (case 2) [9] (1988)	52	♀	*Zooepidemicus*	Consumption of unpasteurized cows’ milk	No	1	No	Mitral	emboli in the arm	Positive	np	PEN G + CTX	4	No	Survive, residual myocardial damage and impairment of peripheral flow in one arm
Edwards et al. (case 6) [9] (1988)	72	♂	*Zooepidemicus*	Consumption of unpasteurized cows’ milk	No	4	No	nr	peripheral embolism	np *	np *	nr	nr	No	Death
Edwards et al. (case 9) [9] (1988)	73	♂	*Zooepidemicus*	Consumption of unpasteurized cows’ milk	ischemic heart disease	5	Meningitis	nr	septic aortic aneurysm	np *	np *	nr	nr	No	Death
Edwards et al. (case 11) [9] (1988)	79	♂	*Zooepidemicus*	Consumption of unpasteurized milk or cheese	No	3	Pneumonia	nr	No	np *	np *	CTX + AMP + MTR	2, then died	No	Death
Martinez-Luengas et al. [40] (1982)	51	♂	*Zooepidemicus*	nr	rheumatic heart disease	1	No	Mitral	petechial hemorrhages right foot	Positive	np	PEN G + STR	4	No	Survive

nr: not reported; np: not performed; * autopsy diagnosis; CCI: Charlson Comorbidity Index; TTE: transthoracic echocardiogram; TOE: transoesophageal echocardiogram; CHD: chronic heart disease; AF: atrial fibrillation; PM: pacemaker; PEN G: penicillin G; AMX: Amoxicillin; AMP: Ampicillin; AMX/CLV: Amoxicillin/Clavulanic acid; AMP/SLB: Ampicillin/Sulbactam; PIP/TZB: Piperacillin/Tazobactam; CRO: ceftriaxone; CTX: cefotaxime; GNT: gentamicin; VAN: Vancomycin; MTR: Metronidazole; LNZ: linezolid; RIF: Rifampicin; STR: Streptomycin.

**Table 2 microorganisms-12-02201-t002:** Patient demographics and infection characteristics of 25 cases of *St. equi* endocarditis described in literature.

Variable	All Patients (n = 25)
Male sex, n (%)	19 (76)
Mean age, years (IQR)	65.0 (57–73)
*St. equi* subspecies, n (%) *Zooepidemicus* *Ruminatorum* Not determinable	22 (88.0) 2 (8.0) 1 (4.0)
Origin of infection, n (%) Contact (direct or undirect) with horses Consumption of unpasteurized milk products Not found Not reported	14 (56.0) 7 (28.0) 3 (12.0) 1 (4.0)
Co-morbidities, n (%) Hypertension Diabetes mellitus Chronic Kidney Disease Oncologic Disease Immunosuppression Dementia Hearth diseases Prosthetic valve Previous valve malformation Ischemic heart disease Mean Charlson Comorbidity Index	9 (36.0) 2 (8.0) 2 (8.0) 1 (4.0) 1 (4.0) 2 (8.0) 12 (48.0) 6 (24.0) 3 (12.0) 3 (12.0) 2.72
Laboratory findings at admission Serum CRP, median (IQR) g/L WBC count, median (IQR) cells × 10^3^/mmc	279.5 (168.1–269.3) 13.5 (10.4–16.5)
Multiple site Infection Patients with multi-site infection, n (%) One site infection Two site infection Three site infection Other sites, in addition to endocarditis Pneumonia Meningitis Endophthalmitis Arthritis Shoulder Knee Spondylodiscitis	16 (64.0) 10 (40.0) 4 (16.0) 2 (8.0) 7 (28.0) 9 (36.0) 2 (8.0) 3 (12.0) 1 (4.0) 2 (8.0) 3 (12.0)
Heart Valve affected Aortic Mitral Not reported	15 (60.0) 6 (24.0) 4 (16.0)
Septic emboli, n (%)	14 (56.0)
Antibiotic Treatment (ABT) Not reported, n (%) Monotherapy, n (%) Combination therapy (>1 ABT, simultaneously)	2 (8.0) 5 (20.0) 18 (72.0)
ABT Agent, n/total prescriptions (%)	
Benzylpenicillin Third-generation cephalosporin Aminopenicillins Rifampicin Gentamicin, in combination with; Benzylpenicillin Third-generation cephalosporin Aminopenicillins	11/41 (26.8) 10/41 (24.3) 5/41 (12.2) 5/41 (12.2) 10/41 (24.3) 5/10 (50.0) 3/10 (30.0) 2/10 (20.0)
Duration of ABT Median (range), weeks <6 weeks, n (%) 6 weeks, n (%) >6 weeks, n (%) Not reported, n (%)	6 (4–16) 8 (32.0) 7 (28.0) 2 (8.0) 8 (42.0)
Valve replacement surgery, n (%)	8 (32.0)
Outcomes Death Survive, with residual injury Survive, without residual injury	5 (20.0) 6 (24.0) 14 (56.0)

Note. Data are no. (%) of patients, unless otherwise indicated. IQR = interquartile range.

## Data Availability

The data presented in this study are available on request from the corresponding author due to privacy restrictions.

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
