# Peer review of "Streptococcus equi Subspecies zooepidemicus Endocarditis and Meningitis in a 62-Year-Old Horse Rider Patient: A Case Report and Literature Review"

_microorganisms, 2024, doi:10.3390/microorganisms12112201_

Round 1

Reviewer 1 Report

Comments and Suggestions for Authors

The authors of the manuscript “Streptococcus equi Subspecies zooepidemicus endocarditis and meningitis in a 62-year-old horse rider patient: A Case Report and Literature Review” present the case of a human infection with Streptococcus equi zooepidemicus and the subsequent treatment. They follow this description with a literature review of 24 other cases of Streptococcus equi infections and present summary statistics of commonalities and differences between them.

The manuscript is not well organized and would benefit from a thorough review. I have included comments below, and I hope the authors can use them to improve their submission.

General

-               I would suggest removing case description and doing a literature  review on the topic of human infections caused by Streptococcus equi subspecies zooepidemicus

-               Grammar and language check is necessary. Most of the issues can be identified with spell-check software. I only point out some mistakes in the comments below, but not all of them.

Specific comments

Introduction

1.          Line 45: Which infections? In animals or humans? Please improve the flow of the text.

2.          Line 52: If there are reported cases of human infection, why do you say it “appears to have the potential to cause zoonotic infections in humans”?

3.          Line 55: You did not introduce the meaning of “IE”.

4.          Lines 48 – 66: It appears that this section is not well connected. The information given should be rephrased and connected for better comprehension.

5.          Introduction: To highlight the issue of your study, you should mention open questions about SEZ and clearly mention the need for your study, as well as your goals.

Material and Methods

6.          Line 76: Please provide more details on the criteria for the exclusion of studies.

7.          Table 1: All abbreviations used in the table (including the header) should be explained in the table caption or in the table notes.

Case Report Description

8.          Line 93: Exchange “man dentist” for “male dentist”. Furthermore, is the occupation of the patient relevant to the case description?

9.          Line 94: Add a comma between fever and cough.

10.    Line 140: You introduce the meaning of CT here but have used the abbreviation earlier in line 102.

Discussion and Literature Review

11.    Line 154: Why do you include “Discussion” in the title of this section if you have a Discussion section later?

12.    Lines 186 - 189: How does this compare to the general population of a similar age as the described patients?

Discussion

13.    Line 207: I suggest moving this statement to the Introduction section to describe your reason for this study.

14.    Lines 219 – 227: Why do you present this information here? What is the connection with your findings?

15.    Line 228: Please correct this sentence, “Nearly all cases of majority of IE cases”.

16.    Line 260: Please rephrase to “the treatment in our literature review aligned with…”.

17.    Line 287: “typical”, not “tipical”.

Comments on the Quality of English Language

   Grammar and language check is necessary. Most of the issues can be identified with spell-check software. I only point out some mistakes in the comments below, but not all of them.

Author Response

General comments

Comment 1: I would suggest removing case description and doing a literature review on the topic of human infections caused by Streptococcus equi subspecies zooepidemicus

Response 1: We think that our case description is still valuable and worth to report, underlying the multi-site and aggressive presentation in humans infection and interesting therapeutic approach.

Comment 2: Grammar and language check is necessary. Most of the issues can be identified with spell-check software.

Response 2: Agree. We have extensively revised all the document.

Specific comments

Introduction

  1. Line 45: which infections? In animals or humans? Please improve the flow of the text .
  2. Line 52: if there are reported cases of human infection, why do you say it “appears to have the potential to cause zoonotic infections in humans”?
  3. Line 55: you did not introduce the meaning of IE
  4. Lines 48 – 66: it appears that this section is not well connected. The information

given should be rephrased and connected for better comprehension

  1. Introduction: to highlight the issue of your study, you should mention open questions about SEZ and clearly mention the need for your study, as well as your goals.

Response:

  1. Agree, we now have specified which infections we are talking about
  2. Thank you for pointing this out , we made a review of the statement
  3. We did a comprehensive revision of all the abbreviations used in the text, as well as other little corrections (ex. Italics in bacteria names).
  4. In response to the comment, we have restructured the introduction for improved clarity and coherence. The revised introduction now begins with a classification of Streptococcus equi subspecies (equi, ruminatorum, and zooepidemicus), providing some key characteristics of each. We then focus on Streptococcus equi zooepidemicus, which is the most significant zoonotic subspecies, highlighting its role in various human diseases, including infective endocarditis. Following this, we define endocarditis and specifically focus on streptococcal infective endocarditis, outlining its clinical relevance. Finally, the introduction concludes with the objective of the case report and literature review. We hope that this reorganization makes the section clearer and more logically structured compared to the previous version.
  5. Thanks, our goals specified in an added paragraph from Line 75 to Line 79.

Material and Methods

  1. Line 76: please provide more details on the criteria for the exclusion of studies
  2. Table 1. All abbreviations used in the table (including the header) should be explained in the table caption or in the table notes.

Response:

  1. Specified which key elements led to exclusion from the review, such as the absence of detailed clinical characteristics, demographic information, therapeutic approach, or patient outcomes.
  2. Yes agree, as mentioned before we have made corrections to the form of the presented text .

Case report description

  1. Line 93: exchange “man dentist” for “male dentist”. Furthermore, is the occupation of the pation relevant to the case description?
  2. Line 94: add a comma between fever and cough.
  3. Line 140: you introduce the meaning of CT here but have used the abbreviation earlier in line 102.
  4. Line 154: why do you include “discussion” in the title of this section if you have a discussion section later?
  5. Line 186 – 189: how does this compare to the general population of a similar age as the described patients?

Response:

  1. We have decided to omit this information as it is not relevant.
  2. Done
  3. We have revised all the abbreviations in the text
  4. Agree
  5. I did not fully understand the comment. Could you please clarify the specific issue or point being raised?

Discussion

  1. Line 207: I suggest moving this statement to the introduction section to describe your reason for this study
  2. Lines 219 – 227: why do you present this information here? What is the connection with your findings?
  3. Line 228: please correct this sentence, “nearly all cases of majority of IE cases”.
  4. Line 260: please rephrase to “the treatment in our literature review aligned with…”.
  5. Line 287: “typical”, not “tipical”.

Response:

  1. As mentioned above, we specified our goals in introduction.
  2. Thank you for your valuable comment. I agree that the paragraph on beta-hemolytic streptococcal endocarditis was out of place in the Discussion section. However, as it is important for understanding the severity of streptococcal endocarditis, I have moved it to the introduction, hoping this will enhance the clarity and coherence of the manuscript.
  3. Done
  4. Done
  5. Done

Reviewer 2 Report

Comments and Suggestions for Authors

Dear authors

Thanks for your work and presentation.

However, some comments should be considered during revision;

  1. Line 28, Streptococcus equi should be followed by its abbreviation “S. equi” and this abbreviation should be mentioned all over the manuscript (lines 71 & 81).
  2. Line 28, Gram not GRAM.
  3. Lines 28-34, references should be included.
  4. Line 38, animal not Animal.
  5. Lines 55 and 57, what are the meanings of abbreviations “IE” and EURO-ENDO”, respectively? The abbreviations should be carefully revised all over the manuscript.
  6. There is a problem in writing the names of bacteria and their style, especially streptococcal!
  7. In Table (1), the period (years from….. to……..) of the survey should be identified, besides, the abbreviations in the table (e.g ICI, TEE, etc.) should be identified under it.
  8. Line 131, why this abbreviation of the drug (LNZ) was indicated?
  9. Line 102, the abbreviation “CT” was mentioned, while in line, a thoraco-abdominal computed tomography “CT” was also mentioned!.
  10. Most of the mentioned references are old.
  11. The conclusion should be summarized without repeated ideas.

Best wishes

Author Response

Comment 1: Line 28, Streptococcus equi should be followed by its abbreviation “S. equi” and this abbreviation should be mentioned all over the manuscript

Response 1: We have extensively revised all the document regarding abbreviations, as well as other formal corrections (eg. italics in bacteria names etc).

Comment 2: Line 28, Gram not GRAM

Response 2: Done

Comment 3: Lines 28-34, references should be included.

Response 3: Thank you for pointing this out. It is just an overview of different S. equi subspecies.

Comment 4: Line 38, Animal not Animal

Response 4: Done

Comment 5: Lines 55 and 57, what are the meanings of abbreviations “IE” and “EURO-ENDO”, respectively? The abbreviations should be carefully revised all over the manuscript

Response 5: As mentioned before we made corrections to the form of the presented text . As far as we know EURO-ENDO is not an abbreviations, but the proper name of the ESC-EORP European Infective Endocarditis Registry

Comment 6: There is a problem in writing the names of bacteria and their style, especially streptococcal

Response 6: I have made the revisions according to your comment. However, I am not entirely sure if the modifications fully address the issue, and I would appreciate further feedback on this matter

Comment 7: In table 1, the period (years from…. to…) of the survey should be identified, besides, the abbreviations in the table (eg ICI, TEE, ) should be identified under it.

Response 7: We have clarified the abbreviations. Regarding the survey period, we included all the publications found during the bibliographic research, we mentioned it at line 82 of the text. 

Comment 8: Line 131, why this abbreviation of the drug (LZN) was indicated?

Response 8: We removed that

Comment 9: Line 102, the abbreviation CT was mentioned, while in line, a thoraco-abdominal computed tomography “CT” was also mentioned

Response 9: all the abbreviations used in the text were revised carefully

Comment 10: Most of the mentioned references are old

Response 10: In response to your Comment regarding some references in the article are old, I would like to clarify that certain older citations are necessary because they refer to historical clinical cases included in the literature review. These cases, some dating back several decades, are essential for understanding the progression and reporting of infective endocarditis over time. However, the discussion section of the manuscript extensively references more recent studies from the past few years. For instance, it includes references to Iversen, K. et al. Partial Oral versus Intravenous Antibiotic Treatment of Endocarditis. N Engl J Med 380, 415–424 (2019) and the updated 2023 ESC Guidelines for the Management of Endocarditis.

Additionally, in last and present version I have incorporated the latest 2023 Duke-International Society for Cardiovascular Infectious Diseases Criteria for Infective Endocarditis, which updates the Modified Duke Criteria, ensuring that the manuscript is aligned with the most current diagnostic standards in the field.

Comment 11: The conclusion should be summarized without repeated ideas

Response 11: Done

Reviewer 3 Report

Comments and Suggestions for Authors

Interesting manuscript describing an infective endocarditis (IE) case and a corresponding literature review on IE cases involving Streptococcus equi subspecies.  Although the manuscript is well-written, the organization appears back to front (relative to the title).

It is suggested that the authors place the case study ahead of the literature review (as two subheadings) in the Materials and Methods (i.e. 2.1 case study, then 2.2 Literature review).  The Discussion section could be Section 4 with concluding remarks as section 5.

Some obvious errors are as follows:

1. Line 19 - Streptococcus equi (italics)

2. Lines 28 & 126 - Gram-positive (named after Gram)

3. Line 30 - hemolytic

4. Line 31 - Lancefield group C.

5. Line 55 - define IE on first use.

6. Line 55 - S. aureus is Staphylococcus aureus. This should be abbreviated to St. aureus subsequently to avoid confusion with Streptococcus (S.).

7. Line 60 - Group C streptococcal

8. Line 93 - man dentist? (please clarify)

9.  Line 99 - mm3 - superscript the number, i.e. mm3. Several of these errors occurs in the manuscript.

10. Line 107 - q12h, q8h - please define what these abbreviations mean.

11. Line 120 - diagnosis

12. Line 130 - b-lactams

13. Line 134 - mg/kg/day?

14. Line 283 - Group C streptococcal

Comments on the Quality of English Language

Several grammatical errors but otherwise fine.

Author Response

Although the manuscript is well written, the organization apprears back to front (relative to the title).

It is suggested that the authors place the case study ahead of the literature review (as two subheadings) in the Materials and Methods (ie 2.1 case study , then 2.2 literature review). The discussion section could be Section 4 with concluding remarks as section 5.

Response: Although I agree with this observation, the manuscript is structured according to the journal's guidelines: "Research manuscript sections: Introduction, Materials and Methods, Results, Discussion, Conclusions (optional)."

Comment 1: line 19 – Streptococcus equi (italics)

Response 1 : Thank you for pointing this out. We have made a careful review of the document, with particular attention to abbreviations and italics for bacteria names.

Comment 2: lines 28 & 126 – Gram -positive (named after Gram)

Response 2: As mentioned before we have correct all these formal errors

Comment 3: line 30  - hemolytic

Response 3: done

Comment 4: line 31 – Lancefield group C

Response 4 : agree

Comment 5: line 55  - define IE on first use

Response 5: Added short definition of IE.

Comment 6: line 55 – S. aureus is Staphylococcus aureus. This should be abbreviated to St. aureus subsequently to avoid confusion with Streptoccocus (S.)

Response 6: I see your point, however the classical abbreviations is S. aureus and we prefer to keep it this way as it is more formally correct

Comment 7:  line 60  - group C streptococcal

Response 7: done

Comment 8: line 93: man dentist? (please clarify)

Response 8: we decided to omit the patient’s occupation as it is not relevant to the case

Comment 9: line 99 – mm3

Response: revised

Comment 10: line 107 – q12h, q8h – please define what these abbreviations mean

Response: Thank you for pointing this out. We have now made these abbreviations explicit in the text

Comment 11: line 120 diagnosis

Comment 12: line 130 b- lactams

Comment 13: line 134 – mg/kg/day?

Comment 14: line 283 – group C streptococcal

Response 11, 12, 13, 14: revised all these corrections, thank you.